# Chronic Inflammation and Immune Dysregulation in Metabolic-Dysfunction-Associated Steatotic Liver Disease Progression: From Steatosis to Hepatocellular Carcinoma

**DOI:** 10.3390/biomedicines13051260

**Published:** 2025-05-21

**Authors:** Young-Min Jee, Jeong-Yoon Lee, Tom Ryu

**Affiliations:** 1Department of Family Medicine, Soonchunhyang University Seoul Hospital, Seoul 04401, Republic of Korea; ymjee27@gmail.com; 2Department of Family Medicine, Graduate School of Medicine, Seoul National University College of Medicine, Seoul 03080, Republic of Korea; 3Department of Neurology, Soonchunhyang University Seoul Hospital, Seoul 04401, Republic of Korea; ljy890818@schmc.ac.kr; 4Department of Translational Medicine, Graduate School of Medicine, Seoul National University College of Medicine, Seoul 03080, Republic of Korea; 5Department of Internal Medicine, Institute for Digestive Research, Digestive Disease Center, Soonchunhyang University College of Medicine, Seoul 04401, Republic of Korea

**Keywords:** metabolic-dysfunction-associated steatotic liver disease, immunity, inflammation, liver fibrosis, hepatocellular carcinoma

## Abstract

**Background/Objectives**: Metabolic-dysfunction-associated steatotic liver disease (MASLD) progresses from hepatic steatosis to hepatocellular carcinoma (HCC) as a result of systemic immunometabolic dysfunction. This review summarizes the key roles of the innate and adaptive immune mechanisms driving hepatic injury, fibrogenesis, and carcinogenesis in MASLD. **Methods**: A comprehensive literature review was performed using PubMed to identify relevant published studies. Eligible articles included original research and clinical studies addressing immunological and metabolic mechanisms in MASLD, as well as emerging therapeutic strategies. **Results**: We highlight the roles of cytokine networks, the gut–liver axis, and immune cell reprogramming. Emerging therapeutic strategies, including cytokine inhibitors, anti-fibrotic agents, metabolic modulators, and nutraceuticals, offer several indications for attenuating MASLD progression and reducing the prevalence of extrahepatic manifestations. **Conclusions**: Given the heterogeneity of MASLD, personalized combination-based approaches targeting both inflammation and metabolic stress are essential for effective disease management and the prevention of systemic complications.

## 1. Introduction

Metabolic dysfunction-associated steatotic liver disease (MASLD) is a recently adopted nomenclature that reflects a paradigm shift in the understanding of fatty liver diseases [1]. Previously referred to as nonalcoholic fatty liver disease (NAFLD), the term was widely used to describe a spectrum of liver disorders characterized by hepatic steatosis in individuals without significant alcohol consumption [2]. However, criticisms of this terminology have accumulated over the past decade. The label “nonalcoholic” defines the disease by what it is not, rather than by what it is, and it fails to acknowledge the key pathogenic role of metabolic dysfunction. Furthermore, the term “fatty” has been considered by many to be stigmatizing, both for patients and for public communication [3].

In response to these concerns, a comprehensive, multi-society Delphi consensus involving experts from hepatology, endocrinology, pediatrics, and public health was conducted under the guidance of major liver associations. As a result, the term MASLD was endorsed as a replacement for NAFLD, and the diagnostic criteria were redefined to require evidence of hepatic steatosis in conjunction with at least one cardiometabolic risk factor (CMRF), such as obesity, type 2 diabetes mellitus, or hypertension. Additionally, metabolic dysfunction-associated steatohepatitis (MASH) has replaced the term NASH to reflect the inflammatory and fibrotic progression of MASLD in affected individuals [1].

This reclassification acknowledges MASLD as a metabolic and systemic disease that can progress from simple steatosis to more advanced stages, including MASH, liver fibrosis, cirrhosis, and eventually hepatocellular carcinoma (HCC) [4]. Central to this progression is the persistent activation of innate and adaptive immune pathways, chronic low-grade inflammation, and cross-talk between the liver and extrahepatic organs such as the gut, adipose tissue, and the immune system [5]. Indeed, inflammation is now recognized not merely as a consequence, but as a driver of disease progression, mediated by hepatokines, cytokines, metabolic stress, and immunological dysregulation [6,7].

Recent epidemiological analyses have highlighted the growing global burden of MASLD, which is now recognized as the most common chronic liver disease, affecting over one-third of adults worldwide. MASLD is now the leading cause of chronic liver disease globally, affecting over 38% of adults and up to 14% of children, although, in regions where viral hepatitis remains prevalent, it continues to require significant public health focus [8]. According to a recent microsimulation study based on United States of America data, the prevalence of MASLD is projected to increase from 33.7% in 2020 to 41.4% by 2050, corresponding to approximately 122 million affected adults. This alarming trajectory highlights the urgent need for preventive and therapeutic strategies at the population level [9]. In addition to liver-specific outcomes such as fibrosis and carcinogenesis, MASLD imposes a significant health and economic burden through its association with cardiovascular disease, chronic kidney disease, and various extrahepatic cancers [10].

In this review, we provide an overview of the immune-mediated mechanisms that underlie the transition from steatosis to advanced liver pathology in MASLD. We revealed the roles of innate and adaptive immune cells, key cytokine networks, and chronic inflammatory signals that contribute to hepatic injury, fibrogenesis, and hepatocarcinogenesis. In addition, we discuss the emerging therapeutic approaches that target these immune pathways, aiming to prevent or reverse MASLD progression.

## 2. Immune Landscape in MASLD

Chronic hepatic inflammation in MASLD is orchestrated by the complex interplay between metabolic signals and the immune system. The dynamic interplay between immunometabolism and epigenetic modulation is increasingly recognized to be a key determinant in the progression of liver diseases, including MASLD, autoimmune hepatitis, and cholestatic disorders [11]. Both innate and adaptive immunity play key roles in hepatic injury, inflammation, fibrogenesis, and hepatocarcinogenesis, beginning early in steatosis and accelerating disease progression.

### 2.1. Innate Immunity in MASLD

Innate immune cells are the first responders to metabolic danger signals, including lipotoxicity, oxidative stress, and gut-derived pathogen-associated molecular patterns (PAMPs). These stimuli activate pattern recognition receptors (PRRs), such as toll-like receptors (TLRs) and nucleotide-binding oligomerization domain-like receptors (NLRs), on both hepatic and non-hepatic immune cells [12,13,14]. Kupffer cells, the liver-resident macrophages, play a central role in sensing free fatty acids, bacterial lipopolysaccharides (LPS), and damage-associated molecular patterns (DAMPs) [15]. Upon activation, Kupffer cells secrete pro-inflammatory cytokines such as tumor necrosis factor-alpha (TNF-α), interleukin (IL)-1β, and IL-6, contributing to hepatocellular damage and the recruitment of additional immune cells [16].

Additionally, neutrophils infiltrate the steatotic liver and promote inflammation by releasing reactive oxygen species (ROS) and neutrophil extracellular traps (NETs). NETs not only damage hepatocytes, but also activate hepatic stellate cells (HSCs), promoting fibrogenesis [17,18]. Monocyte-derived macrophages and dendritic cells (DCs) also accumulate in response to chemokines, further amplifying the local inflammatory environment. DCs, in particular, are involved in antigen presentation and priming of adaptive immune responses, bridging innate and adaptive immunity [19].

Moreover, Kupffer cells tend to polarize toward a pro-inflammatory M1-like phenotype in MASLD, characterized by elevated secretion of TNF-α and IL-1β. This polarization not only sustains hepatic inflammation, but also impairs tissue repair mechanisms associated with the anti-inflammatory M2 phenotype [20].

The contribution of liver sinusoidal endothelial cells (LSECs) is also significant. Although not classical immune cells, LSECs regulate leukocyte recruitment by expressing adhesion molecules such as vascular cell adhesion molecule-1 (VCAM-1) and intercellular adhesion molecule-1 (ICAM-1) in response to inflammation [21]. In MASLD, LSEC dysfunction enhances immune cell infiltration and supports chronic inflammation [22].

In parallel, the recognition of gut-derived LPS by TLR4 on Kupffer cells and HSCs activates the myeloid differentiation primary response 88 (MyD88)-nuclear factor kappa-light-chain-enhancer of activated B cells (NF-κB) signaling cascade, which amplifies inflammatory gene expression and drives fibrosis [12]. This highlights the crucial role of the gut–liver axis in sustaining innate immune activation.

The heterogeneity of liver macrophages, including both resident Kupffer cells and recruited monocyte-derived macrophages, plays a critical role in orchestrating inflammation in MASLD. Recent studies using single-cell technologies has revealed that these populations differ in origin and function, with infiltrating macrophages showing stronger pro-inflammatory and fibrogenic phenotypes [23]. Additionally, neutrophil NETs have emerged as active contributors to hepatic injury and fibrogenesis, linking innate immune activation to long-term disease progression [24].

### 2.2. Adaptive Immunity and Chronic Inflammation

As MASLD progresses, adaptive immune cells become increasingly involved. The liver is enriched with CD4+ T helper (Th) cells, CD8+ cytotoxic T cells, and B lymphocytes, all of which contribute to persistent inflammation and tissue remodeling [25,26].

Among CD4+ T cells, Th1 and Th17 subsets are particularly pathogenic. Th1 cells secrete interferon-gamma, which promotes macrophage activation and enhances cytotoxic responses [27]. Th17 cells, characterized by the production of IL-17A/F, contribute to neutrophil recruitment and liver fibrosis via the activation of HSCs [28]. Notably, IL-17 pathways are upregulated in both murine models and human MASLD, correlating with disease severity [29,30].

CD8+ T cells present cytotoxic effects on hepatocytes, exacerbating cell death and inducing chronic liver injury [31]. While CD8+ T cells initially contribute to host defense, prolonged activation in MASLD may lead to an exhausted phenotype, characterized by the upregulation of inhibitory receptors such as programmed death-1 (PD-1) and T cell immunoglobulin mucin domain-3 (TIM-3), which paradoxically sustains low-grade chronic inflammation [32].

Moreover, regulatory T (Treg) cells, which normally counterbalance inflammation, are often functionally impaired or reduced in number in MASLD, shifting the balance toward immune activation [31]. This disrupted Treg/Th17 balance is a key feature of MASLD progression and is associated with increased fibrosis [25].

B cells and their secreted antibodies may also participate in disease pathogenesis by forming immune complexes and presenting antigens, although their role remains less well defined [33]. Emerging evidence suggests the presence of ectopic lymphoid structures composed of B and T cells in advanced MASLD, indicating a shift toward localized, organized immune responses within the liver [28].

## 3. Chronic Inflammation in MASLD Progression

Chronic inflammation serves as a central driver in the progression of MASLD, orchestrating the transition from benign hepatic steatosis to MASH, liver fibrosis, cirrhosis, and eventually HCC. While simple steatosis is initially reversible, persistent metabolic insults and dysregulated immune responses result in a self-perpetuating cycle of hepatic injury, immune activation, and fibrogenesis. This inflammatory process has recently been conceptualized as a “domino effect”, in which the sensing of stress by innate immune cells triggers the sequential recruitment of pro-inflammatory myeloid and adaptive lymphoid cells, thereby perpetuating hepatic inflammation and fibrosis in a staged manner [33].

### 3.1. Metabolic Stress and Lipotoxicity as Initiating Triggers

The inflammatory cascade in MASLD is primarily initiated by lipotoxicity, resulting from the accumulation of excess free fatty acids and harmful lipid intermediates within hepatocytes [34]. These lipids induce mitochondrial dysfunction, endoplasmic reticulum (ER) stress, and oxidative stress, leading to the excessive generation of ROS. ROS not only damage hepatocyte organelles and DNA, but also amplify inflammatory signaling through redox-sensitive transcription factors such as NF-κB [35,36].

Importantly, unresolved ER stress activates the unfolded protein response, which involves signaling through inositol-requiring enzyme 1, protein kinase R-like ER kinase, and activating transcription factor 6 [37]. Chronic activation of these pathways enhances expression of pro-apoptotic and inflammatory mediators such as C/EBP homologous protein and X-box binding protein 1, thereby exacerbating hepatocellular damage and inflammation [38].

Cell death pathways, including apoptosis, necroptosis, and pyroptosis, release DAMPs such as high-mobility group box 1, adenosine triphosphate, and mitochondrial DNA, which serve as endogenous danger signals [39]. These DAMPs are recognized by PRRs like TLRs (especially TLR4 and TLR9) and NLRs (especially nucleotide-binding domain, leucine-rich-containing family, pyrin domain-containing-3 (NLRP3) inflammasome) on Kupffer cells and DCs, initiating inflammatory responses and cytokine secretion [40]. PAMPs, derived primarily from gut microbiota such as LPS, are recognized by TLR4, triggering pro-inflammatory cascades. DAMPs, such as mitochondrial DNA released from stressed hepatocytes, further activate immune responses through receptors like TLR9 and NLRP3. The NLRP3 inflammasome promotes the cleavage of pro-IL-1β via caspase-1, leading to pyroptosis and enhancing hepatic inflammation and fibrosis [40,41].

### 3.2. Gut–Liver Axis: Microbial Signs and Barrier Dysfunction

In addition to lipotoxic signals, the gut–liver axis plays a crucial role in accelerating hepatic inflammation. In MASLD, intestinal dysbiosis and increased permeability—often referred to as “leaky gut”—facilitate the translocation of microbial-derived PAMPs such as LPS and bacterial DNA into the portal circulation [42]. These PAMPs engage TLR4, TLR9, and other innate immune receptors on Kupffer cells, HSCs, and LSECs, triggering the release of pro-inflammatory cytokines and promoting immune cell recruitment [21,43,44].

Dysbiosis in MASLD is characterized by an altered Firmicutes-to-Bacteroidetes ratio and an enrichment of Gram-negative endotoxin-producing bacteria, such as Enterobacteriaceae [45]. Additionally, levels of beneficial microbial metabolites like butyrate are reduced [46]. This butyrate deficiency impairs Treg differentiation in the gut-associated lymphoid tissue, which could lead to systemic immune imbalance and worsened hepatic inflammation [47]. Recent finding suggested that natural compounds, such as Cornus officinalis vinegar, may ameliorate hepatic steatosis by modulating the gut microbiota and regulating lipid droplet metabolism [48].

The gut microbiota also influences the composition of short-chain fatty acids and secondary bile acids, both of which modulate immune tone and epithelial integrity in MASLD [49]. Disruptions in these microbial metabolites could further disrupt immune responses toward inflammation, exacerbating liver injury.

### 3.3. Immune Activation and Cytokine Networks

Activated Kupffer cells and recruited monocyte-derived macrophages produce a cascade of pro-inflammatory cytokines, including TNF-α, IL-1β, and IL-6. These mediators not only promote hepatocyte damage, but also enhance the recruitment of neutrophils and T cells, enhancing the inflammatory response [50]. Neutrophils contribute to further injury through the release of reactive oxygen species and NETs, which have been shown to activate HSCs and induce fibrogenesis [18].

The C-C motif chemokine ligand (CCL) 2-C-C chemokine receptor (CCR) 2 chemokine axis is particularly important in MASLD, as it drives the recruitment of Ly6C^hi^ monocytes into the liver. Once recruited, these monocytes differentiate into inflammatory macrophages that promote cytokine production and support fibrogenesis [51]. Simultaneously, activation of the NLRP3 inflammasome in macrophages and hepatocytes leads to caspase-1-mediated maturation of IL-1β and IL-18, promoting an escalating inflammatory response. These cytokines stimulate additional immune cell infiltration and sustain a chronic inflammatory environment [52].

Moreover, prolonged activation of NF-κB and c-Jun N-terminal kinase pathways in hepatocytes, immune cells, and HSCs further drive the expression of inflammatory and fibrogenic genes. This signaling axis bridges metabolic stress with immune dysfunction and has been implicated in both steatohepatitis and hepatocarcinogenesis [53]. Beyond cytokines, immunometabolic reprogramming further sustains the inflammatory state. For example, lipotoxic Kupffer cells shift toward glycolysis and exhibit an M1-like pro-inflammatory polarization, which reinforces the production of TNF-α and IL-6 [54].

### 3.4. Adaptive Immune Contribution and Imbalance

In progressive MASLD, the innate immune response transitions into a chronic phase that involves adaptive immune components. CD4+ Th1 and Th17 cells, activated by antigen-presenting DCs and inflammatory cytokines, secrete interferon-gamma and IL-17A, which promote hepatocyte apoptosis and activate HSCs [55]. CD8+ cytotoxic T lymphocytes exacerbate tissue injury through the release of perforin and granzyme, contributing to parenchymal destruction and regeneration stress [56].

Emerging evidence suggests that prolonged antigen stimulation might drive CD8+ T cells toward an exhausted phenotype. While direct studies in MASLD are limited, chronic liver disease models show the upregulation of inhibitory receptors, such as PD-1 and TIM-3 on CD8+ T cells [57,58]. Paradoxically, these exhausted T cells may still contribute to sustained low-grade inflammation through non-cytolytic mechanisms [59].

Tregs, which are essential for maintaining immune tolerance and controlling inflammation, are often reduced or functionally impaired in MASLD, leading to unchecked immune activation. This imbalance between pro-inflammatory and anti-inflammatory T cell subsets is a hallmark of advanced stages of the disease [60]. Notably, the Th17/Treg ratio is a proposed biomarker of disease severity in MASH. Elevated Th17 cytokines (IL-17A/F and IL-22) have been linked to fibrosis progression, whereas Treg dysfunction contributes to the failure of inflammatory resolution [61].

### 3.5. Fibrogenesis and Systemic Inflammatory Crosstalk

Liver fibrogenesis is induced by the activation of HSCs and the deposition of extracellular matrix proteins, including type I and III collagens [62]. Transforming growth factor-beta (TGF-β), platelet-derived growth factor, and IL-13, derived from immune cells and activated hepatocytes, are major profibrogenic cytokines that maintain HSC activation and promote scar formation [63].

Furthermore, MASLD is not restricted to the liver. Chronic inflammation affects systemic circulation, promoting insulin resistance, endothelial dysfunction, and increased risk of comorbidities such as cardiovascular disease and chronic kidney disease [64,65]. Adipose tissue inflammation, marked by macrophage infiltration and altered adipokine profiles, reinforces hepatic immune dysregulation in a bidirectional manner. Additionally, systemic pro-inflammatory cytokines like IL-6 and TNF-α contribute to endothelial activation, increasing the expression of vascular adhesion molecules and promoting atherosclerosis [66]. Similar mechanisms may underlie renal inflammation and fibrosis observed in CKD comorbid with MASLD [67]. Additionally, aging-related immune dysregulation, characterized by the impaired resolution of inflammation and increased pro-inflammatory cytokines, may further exacerbate MASLD progression in elderly individuals with metabolic comorbidities [68].

### 3.6. Dysregulated Innate Immune Pathways in MASLD

The progression of MASLD is importantly driven by dysregulated innate immune signaling pathways. PAMPs, such as LPS derived from the gut microbiota, and DAMPs, including mitochondrial DNA released from injured hepatocytes, are key initiators of hepatic inflammation. These signals are recognized by pattern recognition receptors, particularly TLRs and NLRs, which are expressed on Kupffer cells, HSCs, and other liver-resident immune cells [69]. One of the most critical downstream mediators is the NLRP3 inflammasome. Upon activation by these danger signals, NLRP3 assembles a multiprotein complex that activates caspase-1, leading to the maturation of IL-1β and IL-18. These cytokines enhance hepatic inflammation, promote pyroptotic cell death, and recruit additional immune cells, thereby sustaining a pro-inflammatory and fibrogenic microenvironment [70].

In parallel, the Notch signaling pathway has emerged as a central regulator of hepatic immune remodeling in MASLD. Originally characterized in developmental biology, Notch signaling is now recognized for its role in the activation of hepatic progenitor cells and the modulation of macrophage polarization. Activation of Notch signaling in the liver promotes M2-like macrophage phenotypes and suppresses cytotoxic immune responses, contributing to fibrotic remodeling and the development of an immunosuppressive microenvironment [71]. Altogether, the dysregulation of PAMP/DAMP sensing, inflammasome activation, and developmental signaling pathways such as Notch constitutes a key axis in the immunopathogenesis of MASLD and its progression toward advanced liver disease.

## 4. From Steatosis to HCC: Immune-Modulated Progression

The progression from steatosis to HCC in the context of MASLD is not a linear transition, but rather a dynamic and multi-step process shaped by sustained inflammation, immune dysregulation, fibrotic remodeling, and oncogenic reprogramming [72]. Unlike viral or alcohol-related liver disease, MASLD-derived HCC frequently arises in non-cirrhotic livers, highlighting distinct pathophysiologic mechanisms rooted in metabolic inflammation and altered immune surveillance [73].

### 4.1. Chronic Inflammation and Fibrosis as Pro-Carcinogenic Drivers

Persistent low-grade inflammation in MASLD initiates a cascade of events that predispose hepatocytes to malignant transformation [74]. Kupffer cells, monocyte-derived macrophages, and neutrophils remain chronically activated, releasing ROS, nitrogen intermediates, and pro-inflammatory cytokines such as TNF-α, IL-1β, and IL-6. These mediators not only damage hepatocytes, but also induce compensatory regeneration and genomic instability [75,76].

HSCs are activated concurrently, depositing extracellular matrix proteins that result in progressive fibrosis. Fibrosis, even in the absence of cirrhosis, provides a permissive environment for tumorigenesis by altering tissue architecture, promoting hypoxia, and enabling immune evasion in the context of MASLD [77,78].

### 4.2. Immune Imbalance and Loss of Tumor Surveillance

The immune environment in MASLD has a paradoxical transformation as the disease progresses. While early inflammation is marked by immune activation, later stages are characterized by immune exhaustion and suppression [79]. Chronic T cell stimulation, particularly among CD8+ populations, induces an exhausted phenotype that expresses inhibitory checkpoints like PD-1 and TIM-3 [80]. Tumor-associated macrophages, lipid-associated macrophages, and Tregs further contribute to an immunosuppressive tumor microenvironment [81]. This immune landscape not only fails to eliminate premalignant cells, but may actively support tumor growth by secreting growth factors such as TGF-β and vascular endothelial growth factor.

Importantly, MASLD-related HCC is often associated with the “immune-low” subtype of liver cancer, marked by low T cell infiltration and minimal interferon signaling. This phenotype is thought to be resistant to immune checkpoint blockade therapies, highlighting the necessity for immune-based stratification in MASLD-HCC management [82].

### 4.3. Oxidative Stress, ER Dysfunction, and DNA Damage

Lipotoxicity-driven ROS and reactive nitrogen species induce mitochondrial damage, ER stress, and direct DNA insults in hepatocytes [83]. Accumulation of 8-hydroxy-2′-deoxyguanosine, a biomarker of oxidative DNA damage, has been observed in both human and murine models of MASLD-HCC [84]. These insults impair DNA repair mechanisms, resulting in mutations in tumor suppressors such as tumor protein p53 and activation of proto-oncogenes, including β-catenin and telomerase reverse transcriptase [85]. Unlike traditional cirrhosis-driven HCC, these mutations would emerge earlier in MASLD, prior to architectural distortion, which emphasizes the requirement for early molecular surveillance [77].

Furthermore, dysregulated autophagy and persistent ER stress amplify hepatocellular injury [86]. Proteins such as p62, a component of Mallory–Denk bodies, are prominently expressed in MASLD-HCC, promoting hepatocarcinogenesis through the activation of mammalian target of rapamycin complex 1 and cellular myelocytomatosis oncogene signaling [87].

### 4.4. Oncogenic Signaling Pathways

Multiple oncogenic pathways are activated in MASLD-HCC, often in response to chronic inflammation and metabolic dysfunction. The transcriptional coactivator with PDZ-binding motif (TAZ), which is upregulated by cholesterol accumulation, has been shown to drive hepatocellular transformation by enhancing oxidative stress and upregulating cytochrome B-245 beta chain/nicotinamide adenine dinucleotide phosphate hydrogen oxidase 2-mediated DNA damage [88]. Inhibition of TAZ in murine models suppresses both steatohepatitis and tumor growth, identifying it as a dual-function therapeutic target [89,90].

The wingless-type mouse mammary tumor virus integration site family (Wnt)/β-catenin pathway, often activated through mutations in catenin beta 1, defines a molecular subtype of HCC with immune exclusion and poor prognosis [91]. Notably, this pathway is frequently engaged in MASLD-HCC, contributing to resistance to immune checkpoint therapy [92]. Similarly, signal transducer and activator of transcription 3 (STAT3) signaling is sustained in obesity-related MASLD and HCC, driven by increased IL-6 and leptin. Dysregulated STAT3 activation supports tumor proliferation, survival, and immune evasion [93].

The Notch pathway has recently emerged as a MASLD-specific oncogenic driver. Even in the absence of canonical mutations, Notch signaling is upregulated in HCC arising from MASH livers [94]. The Notch pathway not only supports cholangiocarcinoma-like transcriptional programs, but also modulates immune cell infiltration and macrophage polarization, further supporting an immunosuppressive environment [95]. Evidence suggests that Notch activation contributes to hepatic fibrogenesis and carcinogenesis by influencing macrophage polarization and hepatocyte progenitor expansion, especially in the context of MASLD-associated HCC [96]. Suppression of Notch in experimental models alters the tumor phenotype, but also triggers compensatory Wnt activation, illustrating the complexity of inter-pathway crosstalk [94].

### 4.5. Tumor Evolution and Therapeutic Implications

Post-tumor initiation, the MASLD liver continues to promote tumor evolution through ongoing inflammation, fibrosis, and metabolic dysfunction [97]. The altered microenvironment not only supports malignant progression, but also contributes to therapeutic resistance. Studies have shown that MASLD-HCC patients respond less favorably to immune checkpoint inhibitors compared to those with viral hepatitis-related HCC, possibly due to reduced tumor immunogenicity and the higher prevalence of exhausted CD8+ T cells [79]. An overview of the immunopathogenic mechanisms and the associated immune cell populations contributing to MASLD progression is summarized below (Table 1).

These findings present the need for combinatorial treatment strategies, including agents targeting inflammation including anti-IL-6, anti-STAT3, fibrosis including anti-TGF-β, and metabolism including acetyl-CoA carboxylase (ACC) inhibitor [27,98,99,100]. In this regard, patient-derived xenograft models offer valuable translational insights into MASLD-related hepatocarcinogenesis, providing a platform for evaluating immunometabolic and epigenetic therapies [101]. Novel biomarkers such as patatin-like phospholipase domain-containing protein 3 genotypes and circulating oxidative stress markers might also stratify patients for surveillance and personalized therapy [102].

## 5. Therapeutic Targets and Immune Modulation

The recognition of immune dysregulation and chronic inflammation as central drivers of MASLD progression has catalyzed the development of immune-modulating therapies aimed at interfering with disease progression and improving outcomes [103]. While the traditional management of MASLD has focused on lifestyle modifications and metabolic control, the increasing understanding of the liver’s immune landscape has led to a paradigm shift. Emerging therapies now seek to target key inflammatory mediators, fibrogenic pathways, metabolic stress responses, and the immunosuppressive tumor microenvironment associated with hepatocarcinogenesis. This section outlines current and prospective therapeutic strategies, categorized by their mechanistic focus and clinical potential.

### 5.1. Targeting Pro-Inflammatory Cytokines and Immune Pathways

Chronic inflammation in MASLD is sustained by a network of cytokines and immune cells that amplify hepatic injury and fibrosis. Modulating these immune pathways represents a promising therapeutic option.

IL-6 and the Janus kinase/STAT3 cascade are central to metabolic inflammation. IL-6 can signal through both classical and trans-signaling pathways, the latter being predominant in chronic inflammation [104]. Trans-signaling activates the JAK1/STAT3 axis in hepatocytes, macrophages, and HSCs, promoting survival, proliferation, and fibrogenic gene expression [105]. Monoclonal antibodies targeting IL-6 or IL-6R have demonstrated efficacy in inflammatory diseases and are being explored for MASLD-related immune activation [27]. STAT3 inhibitors, including small molecules like napabucasin and TTI-101, are under investigation for their anti-inflammatory and anti-tumor potential [106,107].

IL-1β and NLRP3 inflammasome signaling play a pivotal role in pyroptosis and the amplification of innate immune responses [108]. Hepatocyte and macrophage-derived DAMPs trigger NLRP3 assembly, leading to caspase-1 activation and maturation of IL-1β and IL-18. These cytokines promote neutrophil recruitment, hepatocyte death, and HSC activation [109]. Nevertheless, neither the IL-1β inhibitor nor NLRP3 inhibitor are in the clinical trials, and urgent evaluation is required.

IL-17 and the Th17/IL-23 axis are increasingly recognized in liver fibrosis and inflammation. IL-17A promotes neutrophil chemotaxis and activates HSCs through the NF-κB and STAT3 pathways [110]. IL-23 maintains the Th17 phenotype, and a dual IL-17/IL-23 blockade is effective in psoriasis and IBD [111,112]. Trials evaluating secukinumab (anti-IL-17A) and risankizumab (anti-IL-23) would provide insights into MASLD applications, particularly in IL-17-high phenotypes.

Immune checkpoint pathways, such as PD-1/PD-L1 and CTLA-4, are essential regulators of immune exhaustion [113]. In MASLD, chronic antigen exposure and metabolic stress induce CD8+ T cell exhaustion and increased checkpoint expression [56]. While checkpoint inhibitors have limited efficacy in MASLD-related HCC due to its “immune-cold” profile, earlier intervention in the MASH stage, particularly in patients with immune infiltration, might be more effective [79]. Combination approaches involving checkpoint blockade and anti-fibrotic or metabolic agents could be the one of the effective options for MASLD-related HCC.

Recent phase II and III trials have demonstrated that agents such as resmetirom [a thyroid hormone receptor (THR)-β agonist], semaglutide (a glucagon-like peptide-1 receptor agonist), and lanifibranor [a pan-peroxisome proliferator-activated receptor (PPAR) agonist] show significant promise in improving the histological features of MASLD, including inflammation resolution and fibrosis reduction. While not yet approved, several of these candidates are in advanced development stages, emphasizing the growing momentum of targeted immunometabolic therapies for MASLD [114].

### 5.2. Anti-Fibrotic Therapies with Immunomodulatory Effects

Fibrosis, driven by sustained immune cell–HSC crosstalk, is the most important predictor of liver-related outcomes in MASLD [27]. As such, targeting fibrogenesis also impacts the hepatic immune environment.

TGF-β signaling is a master pathway in hepatic fibrogenesis [63]. It induces suppressor of mothers against decapentaplegic-dependent transcription of collagen and tissue inhibitor of metalloproteinase, while also presenting immunosuppressive effects [115]. Galunisertib (LY2157299), a TGF-β receptor I kinase inhibitor, has shown anti-fibrotic and anti-tumor activity in an early trial [116]. However, global TGF-β blockade carries risks, including autoimmunity and carcinogenesis, emphasizing the need for tissue-specific or temporally regulated targeting [117].

CCL2–CCR2 and CCL5–CCR5 pathways mediate the recruitment of inflammatory monocytes and macrophages [118]. Cenicriviroc, a dual CCR2/CCR5 antagonist, demonstrated anti-fibrotic effects in the CENTAUR trial, although its phase III results (AURORA) were inconclusive [119,120]. These chemokine pathways also affect immune cell composition, suggesting combinatorial strategies with immunotherapies.

Farnesoid X receptor (FXR) agonists, such as obeticholic acid (OCA), reduce hepatic steatosis, inflammation, and fibrosis. OCA regulates bile acid metabolism, inhibits NF-κB signaling, and reduces chemokine production. However, pruritus and lipid profile changes have limited its broader adoption [121]. Other FXR agonists, such as tropifexor and cilofexor, are in development [122,123].

Fibroblast growth factor (FGF) analogs like FGF19 (Aldafermin) and FGF21 (Pegbelfermin) improve insulin sensitivity, reduce lipotoxicity, and suppress inflammation via PPARγ and adenosine monophosphate-activated kinase activation. These agents modulate hepatic macrophage polarization and might synergize with anti-inflammatory therapies [124].

Lanifibranor, a pan-PPAR agonist, has shown histologic improvement in MASH and fibrosis resolution. PPARα, which activates enhances lipid oxidation, PPARγ, which modulates macrophage activation, and PPARδ, which improves mitochondrial function, would modulate immunometabolic homeostasis together [125].

Recently, THR-β agonists have emerged as promising candidates in MASLD treatment [126]. THR-β, predominantly expressed in hepatocytes, regulates lipid metabolism and mitochondrial function [127]. In MASLD, intrahepatic hypothyroidism, a state characterized by reduced conversion of T4 to T3, is frequently observed and contributes to lipid accumulation, inflammation, and fibrosis [128]. Resmetirom, a selective THR-β agonist, has demonstrated significant reductions in hepatic fat content, serum low-density lipoprotein cholesterol, and markers of fibrosis and inflammation in phase II and III trials [126,129]. It suppresses lipotoxicity, improves deiodinase activity, and downregulates pro-inflammatory pathways including NF-κB and STAT3 [130]. This evidence positions THR-β modulation as a dual metabolic and immune-targeting strategy.

### 5.3. Metabolic Reprogramming and Oxidative Stress Modulation

Hepatic immune cells in MASLD are undergoing metabolic reprogramming, including increased glycolysis and mitochondrial ROS production, which contributes to inflammatory phenotypes [131]. Targeting these metabolic shifts is a novel therapeutic strategy.

ACC inhibitors, such as firsocostat, reduce de novo lipogenesis and substrate overload in hepatocytes and macrophages [100]. In a phase II trial, firsocostat reduced liver fat content and demonstrated trends toward fibrosis improvement [132]. FASN inhibitors (e.g., TVB-2640) inhibit fatty acid synthesis and downstream lipid mediator production, thereby reducing lipid-induced inflammation [133]. Aminocarboxymuconate semialdehyde decarboxylase (ACMSD) inhibition by increasing nicotinamide adenine dinucleotide availability, enhances mitochondrial function, and reduces inflammatory cytokine production [134]. ACMSD inhibitors are also being evaluated for their metabolic and immune benefits in MASLD [135].

Vitamin E, as shown in the PIVENS trial, improved the histologic features of MASH in non-diabetic patients [136]. It reduces oxidative stress, improves hepatocyte survival, and modulates NF-κB activation [137]. However, its long-term safety and efficacy remain under debate [138].

Mitochondria-targeted antioxidants, such as Elamipretide, show potential in reducing hepatocellular ROS and protecting mitochondrial integrity. These agents may also suppress inflammasome activation [139].

### 5.4. Gut–Liver Axis: Microbiota and Immune Crosstalk

The gut microbiota shapes liver immunity through microbial metabolites and endotoxins [140]. Probiotics and synbiotics modulate microbial composition and improve intestinal barrier function [141]. Specific strains such as *Lactobacillus rhamnosus* and *Bifidobacterium longum* have been shown to reduce hepatic inflammation and fibrosis in preclinical models [142]. Clinical studies report improvements in alanine aminotransferase and hepatic steatosis scores with the modulation of gut microbiota [143]. Fecal microbiota transplantation (FMT) has shown benefits in improving gut barrier function and metabolic parameters in pilot MASLD studies [144,145]. However, long-term efficacy, donor variability, and safety concerns limit its use [146]. Postbiotics, such as short-chain fatty acid derivatives and bacterial cell wall fragments, might enhance Treg function and modulate hepatic immunity without requiring live bacteria [147]. This offers a safer and more controllable alternative to FMT.

### 5.5. Nutraceuticals and Adjunctive Immune Modulators

While pharmacologic therapies are evolving, nutraceuticals remain attractive adjuncts due to their safety, accessibility, and potential for systemic immune modulation.

Multivitamin use, particularly antioxidant and B-complex vitamins, has been associated with improved systemic outcomes in MASLD. In a UK Biobank analysis, multivitamin intake correlated with lower cardiovascular and renal risk, although its effects on liver-specific outcomes were modest. These findings suggest that broader antioxidants or micronutrient support may benefit the systemic inflammation and metabolic comorbidities commonly associated with MASLD [148]. In particular, vitamin D deficiency is common in MASLD and correlates with disease severity and insulin resistance [149]. As a modulator of both innate and adaptive immunity, vitamin D enhances regulatory T cell function and suppresses Th17 polarization [150]. Some trials have reported improved liver enzymes and steatosis with supplementation; however, results remain inconsistent, potentially due to differences in baseline levels, dosing, and patient characteristics [151].

Polyphenols, such as resveratrol and curcumin, and omega-3 fatty acids have demonstrated anti-inflammatory, antioxidant, and lipid-lowering properties. These agents act through NF-κB inhibition, the activation of PPARα/γ, and the modulation of gut microbiota composition. In MASLD models, they have been shown to reduce hepatic steatosis and inflammatory cytokine expression [152]. While generally well tolerated, their limited bioavailability and heterogeneous clinical trial designs require more standardized investigations to confirm the efficacy and optimal dosage in MASLD populations.

Glucosamine, traditionally used for osteoarthritis, has also shown potential anti-inflammatory and metabolic benefits relevant to MASLD. A recent large cohort-based study reported that regular glucosamine use was associated with reduced all-cause and liver-related mortality, as well as lower cardiovascular and kidney disease incidence, in patients with MASLD [153]. Although the exact mechanisms remain under investigation, the proposed effects include the inhibition of NF-κB signaling, improvements in insulin sensitivity, and the modulation of oxidative stress [154]. Given its safety profile, glucosamine may be considered to be a complementary approach, particularly in individuals with cardiometabolic comorbidities.

## 6. Conclusions and Future Directions

MASLD is now recognized as a systemic condition characterized by a complex interplay of metabolic stress, immune dysregulation, and chronic low-grade inflammation. As illustrated throughout this review, both innate and adaptive immune responses contribute not only to hepatic injury and fibrogenesis, but also to the progression toward HCC (Figure 1). Inflammatory mediators, immune checkpoints, and tissue remodeling signals shape a dynamic hepatic microenvironment, which can both initiate and sustain disease activity. The immunological mechanisms discussed throughout this review are increasingly supported by a growing body of research, emphasizing the convergence of immune dysregulation, metabolic stress, and fibrosis in MASLD progression. Integrating these insights with emerging translational studies is essential to characterize therapeutic strategies and improve patient outcomes.

Over the past decade, significant advances have been made in unraveling the immunopathogenic mechanisms of MASLD. As summarized in Section 5, multiple compounds, ranging from IL-6 or TGF-β inhibitors to ACC, FASN, and THR-β agonists, are in various stages of clinical development, with some already showing promise in phase II and III trials. Adjunctive interventions, such as nutraceuticals and microbiota-directed therapies, might complement pharmacological approaches by improving systemic metabolic and immune aspects.

Nevertheless, several challenges remain. First, because MASLD varies greatly between individuals, depending on factors like genetics, immune status, other health conditions, and how far the disease has progressed, it is clear that a single, uniform treatment approach would not work for everyone. This highlights the need for more personalized precision-based treatment strategies. Future studies should focus on integrating liver immune phenotyping, non-invasive biomarkers, and genomic risk factors to identify the patient subgroups most likely to benefit from specific interventions. Stratification based on immune activation or suppression states might also help predict responses to immunomodulatory therapies such as checkpoint inhibitors or IL-17 blockers. While several emerging therapies show promise in preclinical or early clinical trials, their real-world applicability remains to be validated. Comprehensive evaluations of long-term safety, adverse events, cost-effectiveness, and population-specific factors are essential before widespread adoption.

Second, while monotherapies have shown promise in selected populations, combination strategies that target multiple pathogenic pathways, such as inflammation, lipotoxicity, and fibrosis, may provide a more effective way of attenuating disease progression or reversing damage. Rationally designed trials that incorporate immune–metabolic synergy and address safety in long-term use are critically required.

Third, the development of non-invasive tools to monitor immune activity and treatment response in real time would be fundamental to guiding therapy. Circulating cytokines, immune cell-derived exosomes, and microbiome-based signatures hold potential as dynamic biomarkers.

Finally, the long-term goal of MASLD management must go beyond improving liver histology to encompass the prevention of extrahepatic complications such as cardiovascular disease, chronic kidney disease, and malignancies. An integrated approach, combining lifestyle intervention, targeted pharmacotherapy, and immune–metabolic optimization, is key to reducing the global burden of this multifaceted disease MASLD progression is highly variable, influenced by genetic susceptibility, lifestyle patterns, and comorbid conditions such as diabetes or cardiovascular disease [155]. These factors require a precision medicine approach tailored to the immunometabolic profile of individual patients.

In conclusion, MASLD represents a unique immunometabolic liver disease that demands novel, integrated, and personalized treatment paradigms. Continued advances in basic immunology, translational research, and biomarker science will enable more effective and individualized care for patients with MASLD.

## Figures and Tables

**Figure 1 biomedicines-13-01260-f001:**
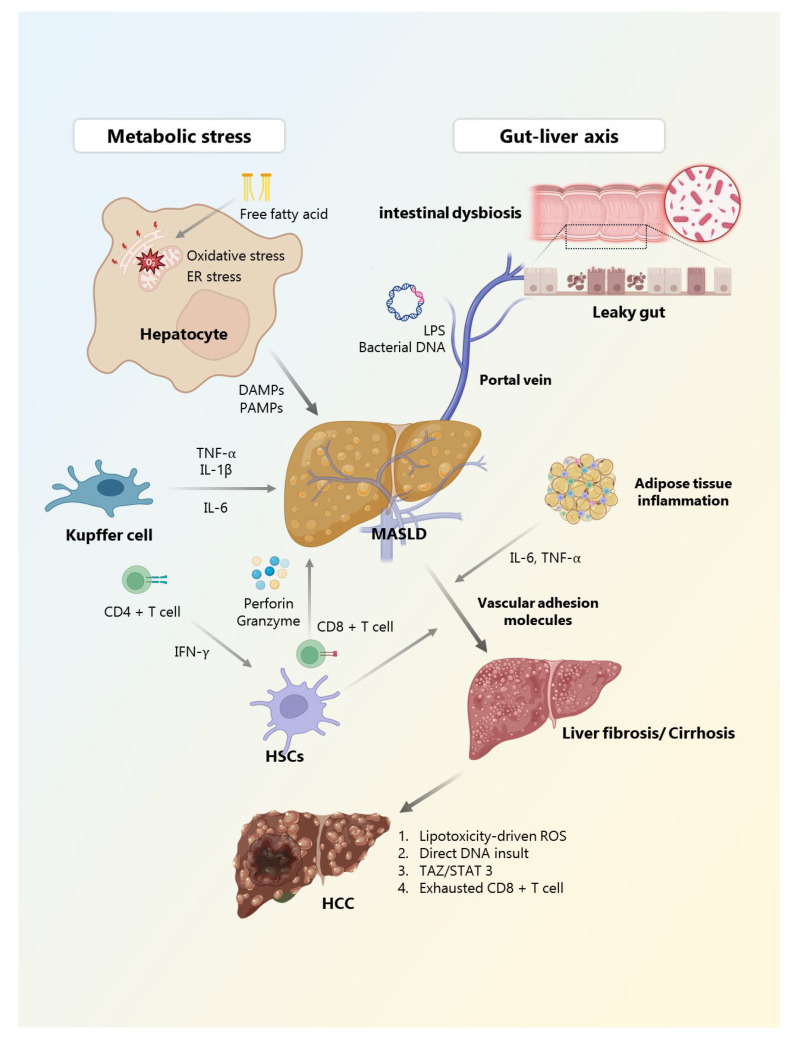
Immune cell interactions and chronic inflammation in MASLD progression to HCC. This figure summarizes the pathophysiological progression of MASLD from simple steatosis to HCC. The figure presents key cellular players, including Kupffer cells, infiltrating macrophages, and T cells, in addition to molecular mediators such as cytokines, DAMPs, and PAMPs, which orchestrate the immunometabolic crosstalk central to MASLD progression. Chronic inflammation and sustained immune dysregulation, driven by metabolic stress, lipotoxicity, and gut–liver axis alterations, contribute to hepatocyte injury, activation of hepatic stellate cells, and progression of liver fibrosis. Over time, these mechanisms interact to promote cirrhosis and malignant transformation. **MASLD**, metabolic-dysfunction-associated steatotic liver disease; **ER**, endoplasmic reticulum; **LPS**, lipopolysaccharide; **DAMPs**, damage-associated molecular patterns; **PAMPs**, pathogen-associated molecular patterns; **TNF-α**, tumor necrosis factor-alpha; **IL**, interleukin; **IFN-γ**, interferon-gamma; **HSC**, hepatic stellate cell; **ROS**, reactive oxygen species; **TAZ**, transcriptional coactivator with PDZ-binding motif; **STAT3**, signal transducer and activator of transcription 3; **HCC**, hepatocellular carcinoma.

**Table 1 biomedicines-13-01260-t001:** Immunopathogenic mechanisms and immune cells in MASLD progression.

Stage/Trigger	Immune Cells	Key Mediators/Pathways	Hepatic Outcome
Lipotoxicity	Kupffer cells [15] DCs [19]	NF-κB [12] ROS [17] DAMPs [15]	Inflammation [20] Immune cell recruitment [22]
Gut-derived signals	Kupffer cells [12] LSECs [21] HSCs [44]	LPS [42] TLR4/TLR9 [43] NLRP3 [14]	Inflammation [47] Fibrosis [44]
Chronic inflammation	Neutrophils [18] Macrophages [51] CD4+/CD8+ T cells [27,32]	IL-6 [66] IL-1β [76] TNF-α [66] NETs [18]	Hepatocyte injury [18] Fibrosis [33]
Adaptive response	Th17 [61] Tregs [60] CD8+ T cells [58]	IL-17A/F [61] IFN-γ [26] PD-1 [58]	Immune imbalance Sustained inflammation [60]
Fibrogenesis, Carcinogenesis	HSCs [77] TAMs [81] exhausted T cells [57]	TGF-β [63] STAT3 [93] Wnt/β-catenin [91]	Fibrosis Immune escape HCC development [79,97]

**DC**, dendritic cells; **NF-κB**, nuclear factor kappa-light-chain-enhancer of activated B cells; **ROS**, reactive oxygen species; **DAMP**, damage-associated molecular patterns; **LSEC**, liver sinusoidal endothelia cell; **HSC**, hepatic stellate cell; **LPS**, lipopolysaccharide; **TLR**, toll-like receptor; **NLRP3**, nucleotide-binding domain, leucine-rich-containing family, pyrin domain-containing-3; **IL**, interleukin; **TNF-α**, tumor necrosis factor-alpha; **NET**, neutrophil extracellular trap; **Th**, T helper; **Treg**, regulatory T cell; **IFN-γ**, interferon-gamma; **PD-1**, programmed death-1; **TAM**, tumor-associated macrophage; **STAT3**, signal transducer and activator of transcription 3; **Wnt**, wingless-type mouse mammary tumor virus integration site family; **HCC**, hepatocellular carcinoma.

## Data Availability

No new data were created or analyzed in this study. Data sharing is not applicable to this article.

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
