# Peer review of "Chronic Inflammation and Immune Dysregulation in Metabolic-Dysfunction-Associated Steatotic Liver Disease Progression: From Steatosis to Hepatocellular Carcinoma"

_biomedicines, 2025, doi:10.3390/biomedicines13051260_

Round 1
Reviewer 1 Report
Comments and Suggestions for Authors
This is a very well-written article. I have a few remarks
- What is the incidence of MASLD worldwide?
- Some dysregulation, such as PAMPs, DAMPs, and NLRP3 activation, may have been deeply described in the context of this pathology. Some pathways are briefly presented, such as Noch.
Author Response
This is a very well-written article. I have a few remarks
- What is the incidence of MASLD worldwide?
Answer: We appreciate this important point. In response, we have incorporated recent global prevalence data in the Introduction section. We now note that MASLD affects over 38% of adults and 14% of children worldwide, reflecting its significant and growing public health burden.
Below is revised Introduction section.
MASLD is now the leading cause of chronic liver disease globally, affecting over 38% of adults and up to 14% of children, although in regions where viral hepatitis remains prevalent, it continues to require significant public health focus (Younossi, Z.M.; Kalligeros, M.; Henry, L. Epidemiology of metabolic dysfunction-associated steatotic liver disease. Clin Mol Hepatol 2025, 31, S32-S50, doi:10.3350/cmh.2024.0431.).
- Some dysregulation, such as PAMPs, DAMPs, and NLRP3 activation, may have been deeply described in the context of this pathology. Some pathways are briefly presented, such as Noch..
Answer: Thank you for this valuable observation. We have expanded our discussion of innate immune dysregulation to provide a more detailed explanation of how PAMPs and DAMPs initiate inflammatory responses via TLRs and NLRP3 inflammasome activation. We also clarified the functional consequences of NLRP3-mediated IL-1β maturation in MASLD progression. Additionally, we elaborated on the Notch signaling pathway in Section 4.4, including its role in hepatic fibrosis and cholangiocarcinoma-like tumor signatures.
Below is revised Section 3 and 4.
PAMPs, derived primarily from gut microbiota such as LPS, are recognized by TLR4, trig-gering pro-inflammatory cascades. DAMPs, such as mitochondrial DNA released from stressed hepatocytes, further activate immune responses through receptors like TLR9 and NLRP3. The NLRP3 inflammasome promotes the cleavage of pro-IL-1β via caspase-1, leading to pyroptosis and enhancing hepatic inflammation and fibrosis (Ma, D.W.; Ha, J.; Yoon, K.S.; Kang, I.; Choi, T.G.; Kim, S.S. Innate Immune System in the Pathogenesis of Non-Alcoholic Fatty Liver Disease. Nutrients 2023, 15, doi:10.3390/nu15092068. , Wree, A.; Eguchi, A.; McGeough, M.D.; Pena, C.A.; Johnson, C.D.; Canbay, A.; Hoffman, H.M.; Feldstein, A.E. NLRP3 inflammasome activation results in hepatocyte pyroptosis, liver inflammation, and fibrosis in mice. Hepatology 2014, 59, 898-910, doi:10.1002/hep.26592.)
The Notch pathway not only supports cholangiocarcinoma-like transcriptional programs but also modulates immune cell infiltration and macrophage polarization, further sup-porting an immunosuppressive environment (Zhu, C.; Ho, Y.J.; Salomao, M.A.; Dapito, D.H.; Bartolome, A.; Schwabe, R.F.; Lee, J.S.; Lowe, S.W.; Pajvani, U.B. Notch activity characterizes a common hepatocellular carcinoma subtype with unique molecular and clinicopathologic features. J Hepatol 2021, 74, 613-626, doi:10.1016/j.jhep.2020.09.032.). An evidence suggests that Notch acti-vation contributes to hepatic fibrogenesis and carcinogenesis by influencing macrophage polarization and hepatocyte progenitor expansion, especially in the context of MASLD-associated HCC (Geisler, F.; Strazzabosco, M. Emerging roles of Notch signaling in liver disease. Hepatology 2015, 61, 382-392, doi:10.1002/hep.27268.).
Reviewer 2 Report
Comments and Suggestions for Authors
This is an excellent, well-structured, and timely review that addresses the critical role of chronic inflammation and immune dysregulation in the pathogenesis and progression of Metabolic dysfunction-associated steatotic liver disease (MASLD) from simple steatosis to hepatocellular carcinoma (HCC). The authors provide a comprehensive overview of the immunopathogenic mechanisms underlying MASLD progression, emphasizing the interplay between innate and adaptive immunity, metabolic stress, gut-liver axis alterations, and fibrogenesis.
The paper aligns with current scientific understanding and reflects recent advances in the field, particularly in light of the updated nomenclature for fatty liver diseases. It also discusses emerging therapeutic strategies , which enhances its clinical relevance.
Author Response
This is an excellent, well-structured, and timely review that addresses the critical role of chronic inflammation and immune dysregulation in the pathogenesis and progression of Metabolic dysfunction-associated steatotic liver disease (MASLD) from simple steatosis to hepatocellular carcinoma (HCC). The authors provide a comprehensive overview of the immunopathogenic mechanisms underlying MASLD progression, emphasizing the interplay between innate and adaptive immunity, metabolic stress, gut-liver axis alterations, and fibrogenesis.
The paper aligns with current scientific understanding and reflects recent advances in the field, particularly in light of the updated nomenclature for fatty liver diseases. It also discusses emerging therapeutic strategies, which enhances its clinical relevance.
Answer: We are grateful for your evaluation and encouraging feedback. We are pleased that the review was recognized for its structure, clarity, and relevance. Your comments motivate us to further characterize our work in line with the evolving landscape of MASLD research and therapeutic development.
Reviewer 3 Report
Comments and Suggestions for Authors
This article primarily explores the progression of MASLD, following its trajectory from hepatic steatosis to HCC, with a particular focus on the implications of systemic immunometabolic dysfunction. It provides a comprehensive summary of the pivotal roles that both innate and adaptive immune mechanisms play in propelling liver injury, fibrosis, and carcinogenesis. The discourse notably emphasizes cytokine networks, the gut-liver axis, and the reprogramming of immune cells. In addition, it delves into emerging therapeutic strategies such as cytokine inhibitors, antifibrotic agents, metabolic modulators, and nutritional supplements, all of which hold promise for slowing down the progression of MASLD and mitigating the incidence of extrahepatic manifestations. The article concludes by asserting that personalized and combination therapies, which target the heterogeneity of MASLD, are essential for effective disease management and the prevention of systemic complications. It is recommended that the author make revisions in accordance with the following suggestions.
- The authors are encouraged to reference and analyze the latest clinical trial data regarding MASLD treatment, thereby bolstering the empirical robustness of the article.
The discourse regarding individual patient differences could be incorporated into the article. 2. This may include consideration of various factors such as genetic backgrounds, lifestyle choices, and comorbidities which may impact the progression of MASLD. The inclusion of these variables could facilitate the development of more personalized treatment plans for patients.
- The following references are highly pertinent to the author's topic and should be appropriately cited by the author.
[1] Cao L, Wu Y, Liu K-Y, et al. Cornus officinalis vinegar alters the gut microbiota, regulating lipid droplet changes in nonalcoholic fatty liver disease model mice. Food & Medicine Homology, 2024, 1(2): 9420002. https://doi.org/10.26599/FMH.2024.9420002
[2] Xu Y-F, Zhao Z-B, Yan EP, Lian Z-X, Zhang W. Complex interplay between the immune system, metabolism, and epigenetic factors in autoimmune liver diseases. Med Adv. 2023; 1(2): 97–114. https://doi.org/10.1002/med4.23
[3] A. Gu, J. Li, M.-Y. Li, Y. Liu, Patient-derived xenograft model in cancer: establishment and applications. MedComm, 2025, 6, e70059. DOI: 10.1002/mco2.70059
- The authors are advised to undertake a comprehensive feasibility analysis of the proposed therapeutic approach, encompassing its real-world applications, potential adverse reactions, and safety considerations.
- The immunological mechanisms addressed in this study can be more effectively integrated with current research endeavors.
Author Response
This article primarily explores the progression of MASLD, following its trajectory from hepatic steatosis to HCC, with a particular focus on the implications of systemic immunometabolic dysfunction. It provides a comprehensive summary of the pivotal roles that both innate and adaptive immune mechanisms play in propelling liver injury, fibrosis, and carcinogenesis. The discourse notably emphasizes cytokine networks, the gut-liver axis, and the reprogramming of immune cells. In addition, it delves into emerging therapeutic strategies such as cytokine inhibitors, antifibrotic agents, metabolic modulators, and nutritional supplements, all of which hold promise for slowing down the progression of MASLD and mitigating the incidence of extrahepatic manifestations. The article concludes by asserting that personalized and combination therapies, which target the heterogeneity of MASLD, are essential for effective disease management and the prevention of systemic complications. It is recommended that the author make revisions in accordance with the following suggestions.
- The authors are encouraged to reference and analyze the latest clinical trial data regarding MASLD treatment, thereby bolstering the empirical robustness of the article.
The discourse regarding individual patient differences could be incorporated into the article.
Answer: Thank you for this insightful suggestion. We have updated Section 5 to include findings from recent phase II and III clinical trials involving promising therapeutic agents such as THR-β agonist, pan-PPAR agonist, and GLP-1 receptor agonist. These agents have shown efficacy in improving NASH resolution, liver histology, and metabolic parameters. We believe this update clarifies the translational relevance of the manuscript.
Below is revised section 5.
Recent phase II and III trials have demonstrated that agents such as resmetirom [a thyroid hormone receptor (THR)-β agonist], semaglutide (a glucagon-like peptide-1 receptor agonist), and lanifibranor [a pan-peroxisome proliferator-activated receptor (PPAR) agonist] show significant promise in improving histological features of MASLD, including inflammation resolution and fibrosis reduction. While not yet approved, several of these candidates are in advanced development stages, emphasizing the growing momentum in targeted immunometabolic therapies for MASLD (Chang, Y.; Jeong, S.W.; Jang, J.Y. Recent updates on pharmacologic therapy in non-alcoholic fatty liver disease. Clin Mol Hepatol 2024, 30, 129-133, doi:10.3350/cmh.2023.0356.).
- This may include consideration of various factors such as genetic backgrounds, lifestyle choices, and comorbidities which may impact the progression of MASLD. The inclusion of these variables could facilitate the development of more personalized treatment plans for patients.
Answer: We fully agree. In the revised Conclusion, we now highlight the heterogeneity of MASLD and discuss how genetic background, lifestyle, and comorbid conditions like type 2 diabetes and cardiovascular disease can influence disease progression and treatment response. We also emphasize the need for precision medicine strategies that tailor interventions to immunometabolic profiles of patients.
Below is revised Conclusions and future directions section.
MASLD progression is highly variable, influenced by genetic susceptibility, lifestyle pat-terns, and comorbid conditions such as diabetes or cardiovascular disease (Zhang, W.; Lu, W.; Jiao, Y.; Li, T.; Wang, H.; Wan, C. Identifying disease progression biomarkers in metabolic associated steatotic liver disease (MASLD) through weighted gene co-expression network analysis and machine learning. J Transl Med 2025, 23, 472, doi:10.1186/s12967-025-06490-7.). These factors require a precision medicine approach tailored to the immunometabolic profile of individual patients.
- The following references are highly pertinent to the author's topic and should be appropriately cited by the author.
[1] Cao L, Wu Y, Liu K-Y, et al. Cornus officinalis vinegar alters the gut microbiota, regulating lipid droplet changes in nonalcoholic fatty liver disease model mice. Food & Medicine Homology, 2024, 1(2): 9420002. https://doi.org/10.26599/FMH.2024.9420002
[2] Xu Y-F, Zhao Z-B, Yan EP, Lian Z-X, Zhang W. Complex interplay between the immune system, metabolism, and epigenetic factors in autoimmune liver diseases. Med Adv. 2023; 1(2): 97–114. https://doi.org/10.1002/med4.23
[3] A. Gu, J. Li, M.-Y. Li, Y. Liu, Patient-derived xenograft model in cancer: establishment and applications. MedComm, 2025, 6, e70059. DOI: 10.1002/mco2.70059
Answer: We appreciate the recommended references and have reviewed them carefully. We have integrated them in the appropriate sections, respectively.
Below is revised section 2, 3, and 4
The dynamic interplay between immunometabolism and epigenetic modulation is increasingly recognized as a key determinant in the progression of liver diseases, including MASLD, autoimmune hepatitis, and cholestatic disorders (Xu, Y.F.; Zhao, Z.B.; Yan, E.P.; Lian, Z.X.; Zhang, W. Complex interplay between the immune system, metabolism, and epigenetic factors in autoimmune liver diseases. Medicine Advances 2023, 1, 97-114.).
Recent finding suggested that natural compounds such as Cornus officinalis vinegar may ameliorate hepatic steatosis by modulating the gut microbiota and regulating lipid droplet metabolism (Cao, L.; Wu, Y.; Liu, K.-Y.; Qi, N.-X.; Zhang, J.; Tie, S.-S.; Li, X.; Tian, P.-P.; Gu, S.-B. Cornus officinalis vinegar alters the gut microbiota, regulating lipid droplet changes in nonalcoholic fatty liver disease model mice. Food & Medicine Homology 2024, 1.).
In this regard, patient-derived xenograft models offer valuable translational insight into MASLD-related hepatocarcinogenesis, providing a platform for evaluating immunomet-abolic and epigenetic therapies (Gu, A.; Li, J.; Li, M.Y.; Liu, Y. Patient-derived xenograft model in cancer: establishment and applications. MedComm (2020) 2025, 6, e70059, doi:10.1002/mco2.70059.).
- The authors are advised to undertake a comprehensive feasibility analysis of the proposed therapeutic approach, encompassing its real-world applications, potential adverse reactions, and safety considerations.
Answer: We appreciate this important reminder. In response, we added a paragraph at the Conclusion section that discusses real-world considerations, including long-term safety, adverse effects, tolerability, and cost-effectiveness. We also note the importance of population-specific validation and post-approval surveillance in translating these therapies into clinical practice.
Below is revised Conclusions and future directions section.
While several emerging therapies show promise in preclinical or early clinical trials, their real-world applicability remains to be validated. Comprehensive evaluation of long-term safety, adverse events, cost-effectiveness, and population-specific factors is essential before widespread adoption.
- The immunological mechanisms addressed in this study can be more effectively integrated with current research endeavors.
Answer: Thank you for this helpful suggestion. To address it, we have inserted new sentences in the Conclusion section that connects the discussed immunological pathways to emerging translational research.
Below is revised Conclusions and future directions section.
The immunological mechanisms discussed throughout this review are increasingly sup-ported by a growing body of research, emphasizing the convergence of immune dysregulation, metabolic stress, and fibrosis in MASLD progression. Integrating these insights with emerging translational studies would be essential to characterize therapeutic strategies and improve patient outcomes.
Round 2
Reviewer 3 Report
Comments and Suggestions for Authors
ACCEPT
Author Response
Reviewer's comment: ACCEPT
Answer: We sincerely thank the reviewer for their positive evaluation and kind support of our manuscript. We are grateful for the encouraging comments.